# Influence of Pyrolysis Temperature on Biochar Produced from Lignin–Rich Biorefinery Residue

**Corinna Maria Grottola \*** , **Paola Giudicianni, Fernando Stanzione and Raffaele Ragucci**

Institute of Sciences and Technologies for Sustainable Energy and Mobility (STEMS) of the National Research Council (CNR), 80125 Naples, Italy

**\*** Correspondence: corinnamaria.grottola@stems.cnr.it

**Abstract:** The biorefinery concept is growing rapidly for bio-based production of fuels and products, and steam explosion is by far the most applied pre-treatment technology allowing the delignification of lignocellulosic biomass. Within the bioethanol production process, pyrolysis of lignin-rich residue (LRR), for producing char to be used in a wide variety of applications, presents a viable way to recover materials and energy, helping to improve the sustainability of the whole production chain. In the present study, it is shown that yields, elemental composition and porosity characteristics of LLR-char are significantly different from those of char produced from alkali lignin. Both products yields and char composition were more similar to the typical values of woody and herbaceous biomasses. The chemical characterization of the chars' organic matrices as well as the content of the main inorganic species suggest the opportunity to perform pyrolysis at low temperatures for producing high yields of chars suitable to be used as carbon sink or soil fertilizers. The BET values of the chars obtained at final temperatures in the range 500–700 °C seem to be promising for char-application processes involving surface phenomena (e.g., adsorption, catalyst support), thus encouraging further analyses of char-surface chemistry.

**Keywords:** lignin; slow pyrolysis; torrefaction; char; biorefinery residues

## 1. Introduction

Europe is committed to have a bio-based economy in 2030 [1]. It follows that a huge contribution of biorefinery products to the European demand for chemicals, energy, materials and fibers is expected in the near future. To be environmentally and economically sustainable, biorefineries will need to be flexible, versatile, energy and cost efficient [2]. Typically, the lignin-rich residue (LRR) separated after sugars fermentation and ethanol distillation, in a biochemical plant aimed at producing bio-ethanol from lignocellulosic feedstock, is used as energy source to support the process thermal needs. However, this potentially available energy is currently only utilized in a small part, less than 2% [3]. In addition, about 60% more lignin is generated than what would be needed to meet the internal energy use [3–5]. The exploitation of this residue for the combined production of biofuels and added-value chemicals and materials represents a key factor for the increase in the efficiency of the overall ethanol production chain. Lignin valorization is mandatory for the viability of future biorefinery operations to alleviate environmental burdens and energy demand strain.

Pyrolysis could be a good candidate for converting the LRR into a solid residue (char), suitable for application in several fields [6], as well as liquid (bio-oil) and gaseous products that can be exploited for energy production or as chemical sources. By properly tuning the main process variables (temperature, heating rate, carrier gas flow rate), the pyrolysis process can be guided towards the modulation of the yields of product fractions and/or the optimization of their chemical and/or physical properties. To this aim, a comprehensive knowledge of the thermal behavior of the LRR residue is needed. A

point often overlooked, pyrolysis also represents a crucial stage in all thermochemical conversions, and it is not only a thermal process for biomass treatment [7]. Therefore, the understanding of lignin-pyrolysis behavior can help to improve the lignin-mechanism decomposition, and is fundamental for effective lignin valorization by a pyrolysis-based biorefinery approach [8].

Fast pyrolysis studies of lignin residues were conducted mainly with the aim of producing added-value chemicals (benzene, toluene, xylene and phenols) for food, chemical and pharmaceutical industry applications [9].

Several authors addressed the understanding of pyrolysis mechanism of, so called, "technical lignin" that is a residue of bioethanol production and pulping. In contrast, few works focused the attention on the progressive evolution of biochar, produced from this feedstock, at different stages of the pyrolytic process [3,10–13].

Pyrolysis of alkali lignin extracted from black liquor in papermaking industry was explored in the temperature range 200–600 °C in the study of [11]. They adopted a two-dimensional perturbation correlation infrared spectroscopy method (2D-PCIS) to study biochar-structural evolution, associated with the properties of the volatile fraction during the thermal treatment. At low temperature, they observed the depolymerization of benzene rings and the breaking/reconstruction of aliphatic chains. These reactions led to the cleavage and devolatilization of simple substituted aryl structures.

In contrast, as temperature increases, the single benzene rings in char evolved into $2 \times 2$ and multi fused-ring structures, corresponding to the enhancement in aromatic ring substitution and polycondensation reactions. In the same temperature range, recombined volatiles forms deposits of fused ring coke on the surface of solid char [11].

Zheng et al., (2022) explored char evolution from pyrolysis of dealkaline lignin in a wider temperature range from 200 to 800 °C through the study of FTIR and Raman spectra. Even at low temperature, they observed the transition of small aromatic rings to more order rings followed by the growth of aromatic monomers leading to the graphitization of the char structure at higher temperature as at 800 °C. However, the major changes of char structure (reduction in carbonyl functionalities and of aliphatic groups) occurred in a narrow temperature region from 520 to 530 °C [13].

As for the investigation on different types of lignin, the international PyNe task of the IEA Bioenergy Agreement Pyrolysis carried out a collaborative study on two different lignin feedstock in fast pyrolysis condition using a fluidized-bed pyrolysis system focused on the produced bio-oil [14]. The undertaken work concluded that the lignin from hydrolysis of softwood, estimated at about 50% lignin and 50% cellulose, behaves like a typical biomass, producing a slightly reduced amount of a fairly typical bio-oil. The lignin from a co-product of manufacture of pulp for printing and writing papers was difficult to process in the fast pyrolysis reactors, due to the scarce production of a very tarry bio-oil with respect to a typical biomass. However, the production of specific chemicals requires both extraction procedures capable of achieving at the same time the multiple objectives of high purity, high yield and, low condensation and the availability of more effective and expensive catalysts to promote selective degradation/conversion routes. In this framework, slow pyrolysis can be considered an economic and robust alternative way of disposing LRR, while producing, without any pretreatment, a carbon based material, biochar, that can be used in a wide variety of applications. Differently from the production of a specific chemical, biochar production is more flexible with respect to the feedstock and the pyrolysis conditions. Some applications are less sensitive to small variations of biochar chemical composition and structure, thus allowing a certain relaxation of the feedstock purity constrain and of the severe control of the production operating conditions. Ghysels et al. [10], for example, assessing the potential of lignin-rich digested stillage in producing biochar for soil amendments, showed that even though heating rate, pyrolysis temperature, and solid holding time at the maximum temperature affected significantly biochar yields, only one or two of these variables affected the chemical characteristics relevant for the agronomic application (fixed carbon, H/C and O/C ratio).

To better understand the potential of the char obtained from the LRR, it is useful to expand the knowledge of its chemical and physical characteristics and their dependence on the pyrolysis conditions.

In this work, a preliminary study on the thermal behavior of the LRR under nitrogen atmosphere was conducted. LRR was obtained from a bio-ethanol, industrial-production plant after distillation of ethanol produced through steam explosion pre-treatment, enzymatic hydrolysis and fermentation. Thermogravimetric analysis and pyrolysis tests were performed at 5 °C/min exploring a wide range of final temperatures from 300 to 700 °C. The pyrolysis products' yields (gas, liquid and char) were quantified and the evolution of the gaseous products during the process was monitored. Special attention was given to the chemical and physical characteristics of the chars, in order to position these materials into the better known landscape of chars derived from untreated vegetal biomasses. To this aim, chemical and physical analyses of chars produced at different temperatures were performed to investigate changes in both the bulk and surface composition. Comparisons with the available data of chars from alkali lignin, and some woody and non-woody residues processed in the same pyrolysis unit under similar operating conditions [15] were provided.

## 2. Materials and Methods

### 2.1. Experimental Reactor

The LRR was pyrolyzed in a laboratory-scale pyrolysis reactor operated under slow pyrolysis conditions (heating rate = 5 °C/min) up to five final temperatures in the range 300–700 °C. The pyrolysis reactor described in [15] consists of a prismatic jacketed chamber (L = 0.024 m, W = 0.04 m, H = 0.052 m) into which in total 6 g of sample are loaded. The longitudinal section of the reactor is sketched in Figure 1.

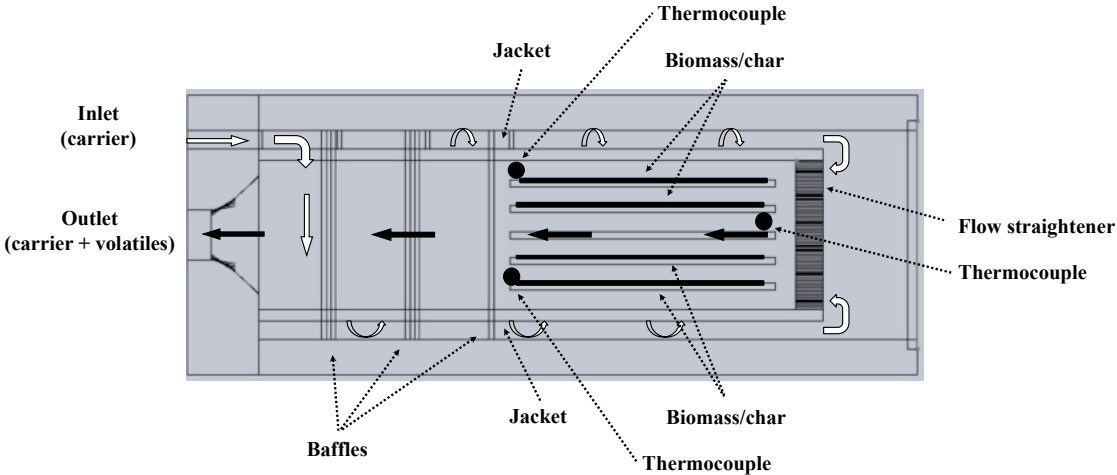

**Figure 1.** Pyrolysis reactor: Cross-section of the reaction chamber.

The sample holder comprises of five trays allocated uniformly along the rectangular cross section of the inner reaction chamber, where biomass is loaded in thin layers (approximately 1 mm thick). A superheater placed before the jacketed reactor heats the carrier gas to the programmed temperature with the aid of a PID controller. The carrier gas flows into the jacket at a constant gas flow rate (3.0 N L/min). Then, the flow is reversed towards the pyrolysis chamber, passing through a ceramic flow straightener before entering the reaction chamber. The absence of relevant thermal gradient inside the pyrolysis reactor during the pyrolysis experiments was assessed by monitoring the temperature through three N-type thermocouples installed in two cross sections at the beginning and end of the sample trays, as shown in Figure 1. At the end of each experiment, the char yield was determined gravimetrically, with respect to the fed sample, using a MS105DU (Mettler Toledo) laboratory balance with 0.01 mg resolution. The gaseous products exiting the reactor flow through a closed-loop forced liquid cooling system maintained at 5 °C and

then through a glass-flask submerged in a dewar filled with liquid nitrogen at $-196$ °C, where the condensable species are collected. The non-condensable gases were sampled and analyzed by means of a micro gas chromatograph, equipped with a thermal conductivity detector (Agilent 3000 Quad), every 171.5 s. Temporal profiles of the release rates of the detected gaseous species ($CO$, $CO_2$, $H_2$, $CH_4$, $C_2H_4$, $C_2H_6$ and $N_2$) were obtained by continuously measuring the carrier gas flow rate and by determining the produced gas composition. The yields of the gaseous products were calculated by integrating the measured release rate curves along the experiment duration. Maximum relative errors for $CO$ $CO_2$, $CH_4$ and $H_2$ species were about $\pm3\%$ (molar fraction). The liquid yields were evaluated as the amount needed to complete the mass balance. Two replicates of each experiment were conducted and a maximum relative error lower than 0.5% has been recorded for all the measured products yield.

*2.2. LRR and Char Characterization Procedures*

The LRR was obtained from a second-generation, bio-ethanol plant fed with an Arundo donax after ethanol distillation. The lignin content was determined according to the procedure described in [16] on an extractives-free sample obtained according to the NREL/TP-510-42619 procedure.

Thermogravimetric analyses of LRR were performed in a thermogravimetric (TG) apparatus (Perkin-Elmer STA6000, Milan, Italy) by heating each sample (~10 mg) at atmospheric pressure under $N_2$ (40 mL/min) from 50 °C up to 700 °C applying a heating rate of 5 °C/min. These operating conditions are similar to those that have been adopted in the subsequent pyrolysis tests.

The LRR samples and the chars produced at different temperatures were characterized through proximate analysis using TGA701 LECO (Milan, Italy) using ASTM E870 procedure. Ultimate analysis (carbon, hydrogen and nitrogen content) was performed with an elemental analyzer CHN 2000 LECO analyzer (Milan, Italy), using EDTA as a standard based on CEN/TS 15104. The oxygen was obtained by difference, considering the measured C, H, N and ash content calculated on dry basis (db).

The content of major inorganic elements was determined by dissolving the LRR and char samples via microwave-assisted acid digestion based on US-EPA Methods 3051 and 3052. A representative sample (200 mg) of biomass was dissolved in 10 mL 65% nitric acid and 1.5 mL $H_2O_2$. The vessel was sealed and heated in the microwave unit at 140 °C for 10 min, then at 180 °C for 30 min (maximum power 1000 W). After they were cooled, the digested samples were analyzed by inductively coupled plasma mass spectrometry (ICP/MS) using an Agilent 7500CE instrument (Milan, Italy).

LRR and char pH was measured with a digital pH meter (827 pH LAB, Metrohm, Milan, Italy) in deionized water using a 1:20 wt./wt. ratio following the ASTM D4972-13 standard procedure.

Finally, the thermal evolution of the chars' structure was followed through Scanning Electron Microscopy (SEM) analysis and by applying gas-adsorption porosimetry. Adsorption/desorption isotherms were obtained using $N_2$ at $-196$ °C as the adsorbate in an Autosorb-1 (Quantachrome) apparatus. Before analysis, the samples were degassed at 200 °C for 6 h under vacuum conditions. The surface area was evaluated using the BET equation. The determination of micropore volume and area was performed using the t-plot method. The total pore volume of the samples was estimated from a single $N_2$ adsorbed point at a relative pressure ($P/P_0$) of 0.95. The mesopore volume was calculated using the BJH pore size distribution theory.

Three replicates were conducted for all the analysis and characterization. The mean values of the three replicates is presented in the Tables and Figures.

## 3. Results

### 3.1. LRR Characterization

The thermal behavior of LRR was studied by following the evolution of the weight loss and its derivative as function of temperature in a TG apparatus. The TG and DTG curves are compared in Figure 2 with the corresponding curves of alkali lignin [15] and of Arundo donax [17], in order to highlight similarities and differences with a commercial pure lignin and the LRR parent feedstock.

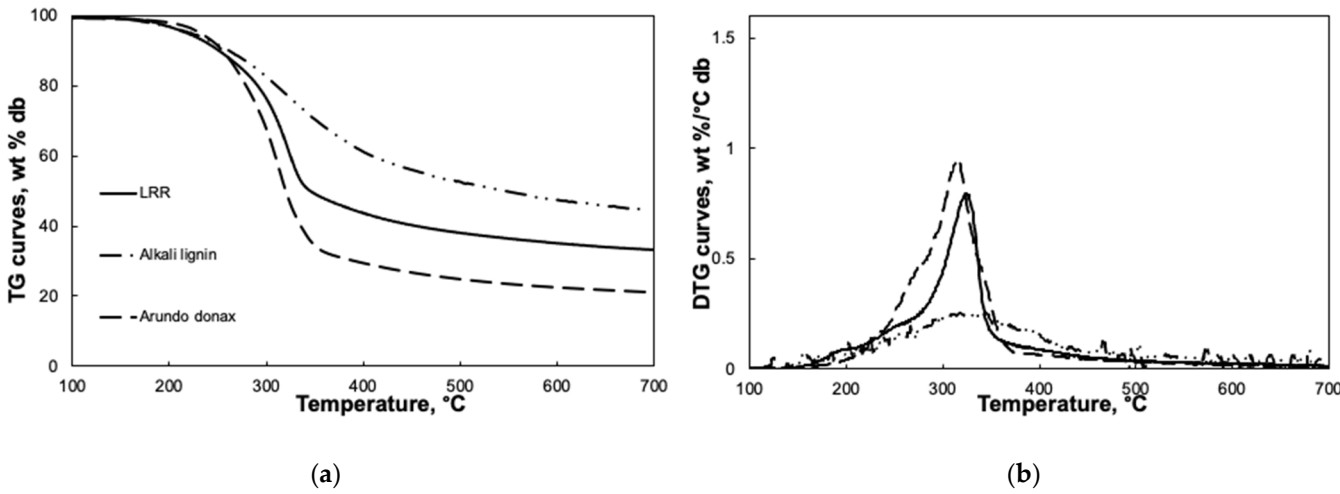

(**a**)                    (**b**)

**Figure 2.** Weight loss and derivative curves as function of temperature of lignin-rich residue (LRR), Alkali lignin and Arundo Donax. (**a**) Thermogravimetric analysis (TG); (**b**) Differential thermal analysis (DTG). For LRR the relative error is lower than 0.5%.

LRR devolatilization starts at about 150 °C as it occurs also for alkali lignin and Arundo donax. However, whereas the devolatilization of alkali lignin proceeds gradually up to 700 °C and it is represented by a smooth, broad peak in the DTG curve, LRR is subjected to a prompt weight loss in the range 280–340 °C represented by a narrow peak in the DTG curve typically attributed to cellulose decomposition. A similar trend is observed for Arundo donax DTG curve. Moreover, on the increasing branch of the DTG curve, a shoulder, typically ascribed to hemicellulose devolatilization, is weakly visible and resembles the corresponding more evident shoulder in the Arundo donax DTG curve. At temperature higher than 340 °C the devolatilization proceeds smoothly with a rate comparable to the one of Arundo donax and lower than the one of alkali lignin. It can be deduced that LRR still contains carbohydrates from the holocellulosic component of the parent feedstock, mainly from cellulose, but it is enriched in lignin, thus resulting in remarkable yield of solid residue at 700 °C.

This observation is consistent with the results of the elemental, proximate and compositional analyses reported in Table 1. The LRR carbon and lignin content is higher than the one of the parent feedstock [18] but a relevant fraction of components other than lignin is still present in the LRR sample. It should be noted that the fixed carbon content of LRR is comparable to the one of Arundo donax [19] and significantly lower than the one expected on the basis of the alkali lignin characterization [15], whereas a higher ash content is obtained for LRR.

As evident from Table 2, ash is mainly composed of K and Ca, consistently with the ash composition of the parent feedstock [19] and totally different form the ash composition of the alkali lignin containing mainly non-inherent Na ascribable to the extraction procedure. It is likely that most of the inorganics are retained in the solid phase during devolatilization up to 700 °C, consequently the relevant amount of solid residue obtained from TG analysis at 700 °C is the result of both the enrichment in lignin and the high ash initial content.

**Table 1.** Elemental and proximate analysis of lignin-rich residue (LRR); alkali lignin [17]; Arundo donax (AD) expressed as wt.% [18]. Percentage error is reported in brackets.

| | C | H | N | O | Volatiles | Fixed Carbon | Ash | Extractives |
|---|---|---|---|---|---|---|---|---|
| | | | Wt. % db | | | Wt. % db | | |
| LRR | 47.3 (0.1) | 5.9 (0.2) | 0.7 (2.3) | 38.2 (0.2) | 69.6 (0.3) | 22.4 (0.5) | 8.0 (0.1) | 31.8 (2.3) |
| Alkali lignin | 65 | 6 | - | 29 | 61.7 | 35.8 | 2.5 | - |
| AD | 43 | 6.1 | - | 45.8 | 75.4 | 23.6 | 1 | 6.5 |

**Table 2.** ICP-MS analysis an pH of lignin rich residue (LRR); alkali lignin [17]; Arundo donax (AD) [18].

| | K | Na | Mg | P | Ca | Fe | pH |
|---|---|---|---|---|---|---|---|
| | | | | mg/kg | | | |
| LRR | 6875 (0.2) | 142 (1.2) | 302 (0.8) | 803 (0.5) | 1791 (0.3) | 448 (0.8) | 5.9 (0.3) |
| Alkali lignin | 657 | 5864 | 118 | 6 | 82 | 38 | - |
| AD | 6378 | 24 | 223 | 364 | 614 | 33 | - |

### 3.2. Pyrolysis Tests

### 3.2.1. Product Yields and Gas Release

The yields of the pyrolysis products are reported in Figure 3. At a temperature higher than 300 °C, the liquid product is the most abundant, consistent with the pyrolytic behavior of the majority of the biomasses [20] On the contrary, Ferreiro et al. [15] showed that alkali lignin pyrolyzed under the same experimental conditions used in this study produced comparable amounts of liquid and char, even at the highest temperature explored in this study. The different behavior of LRR with respect to alkali lignin is due to the presence of holocellulose in the residue responsible of the enhanced production of liquid species.

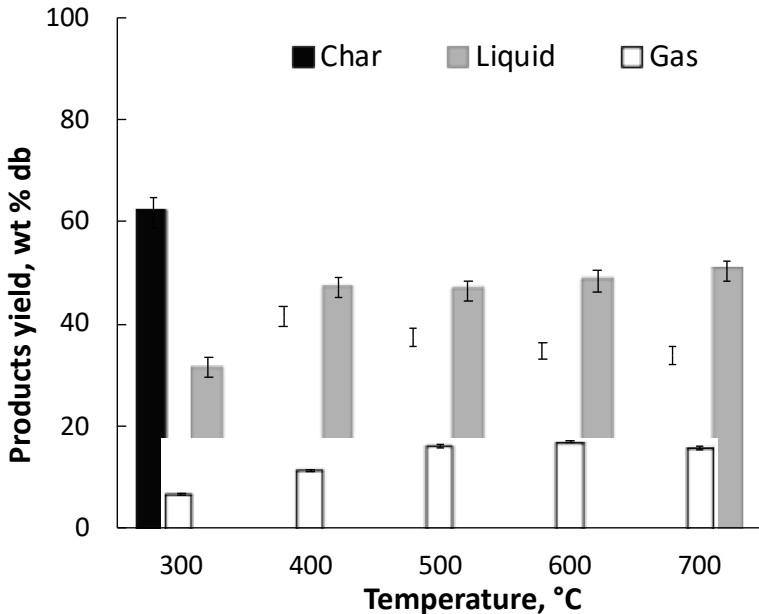

**Figure 3.** Yields of pyrolysis products of lignin rich residue (LRR), at different temperatures.

The decrease in the char yield at increasing pyrolysis temperature follows a similar trend as observed in the TG experiments. A great reduction in the char yields is observed up to 400 °C due to the devolatilization of the residual holocellulose fraction, whereas the reduction in the char yield at higher temperature is sensible. Conversely, liquid-product yield slightly increases with increasing temperature. Differently, the production of gases

increase up to 500 °C and does not vary significantly at higher temperatures. Surprisingly, both the product yields and the gas concentration obtained from LRR in the temperature range 300–700 °C do not correspond to those expected in consideration of the high amount of lignin determined in the LRR sample (c.f. Table 1). In particular, char and liquid yields are, respectively, lower and higher than the ones typical of lignin pyrolysis. At 700 °C, they are rather comparable to the ones obtained from several biomasses processed in the same apparatus and experimental conditions of the present study [15]. A possible reason for this behavior can be found in the analysis of the DTG curve of the LRR sample (see Figure 2). The absence of the typical shoulder representative of the decomposition of hemicellulose on the growing branch of the peak in the range 280–340 °C allows us to deduce that the treatment used for the production of ethanol has led to the hydrolysis of hemicellulose, leaving in the residue, in addition to lignin, mainly cellulose. This assumption is confirmed by the temperature trend of the CO and $CO_2$ release rate curves reported in Figure 4.

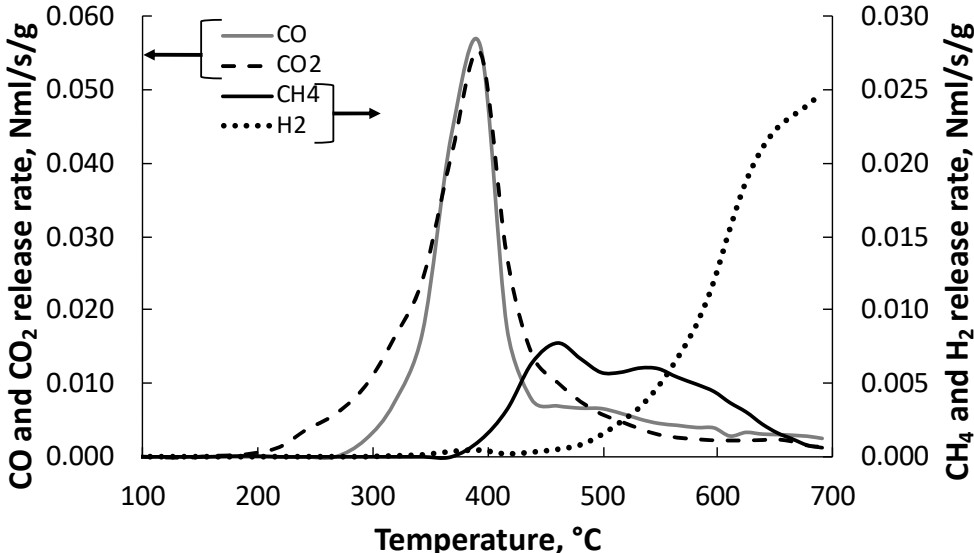

**Figure 4.** Releasing rate (N mL s$^{-1}$ g$^{-1}$ LRR on dry basis) of gaseous species from lignin rich residue (LRR) as function of temperature.

CO and $CO_2$ are released mostly between 300 and 450 °C except for a slight $CO_2$ release at lower temperatures. As reported from the past literature, cellulose decomposes very quickly at temperatures between 300 and 400 °C, and so, $CO_2$ and CO are the main gaseous species released in this temperature region [15]. The same species are released from hemicellulose in a lower temperature region (200–330 °C) [15]. This confirms that hemicellulose has been degraded during the steam explosion and the enzymatic hydrolysis pretreatment in the bioethanol production process. It is known from the literature that cellulose, contrary to lignin, produces high liquid yields and very small char yields [15]. Therefore, the high tendency of lignin to produce char has been compensated by the low tendency of cellulose to do the same. The opposite has occurred for the production of liquids.

Gas composition, reported in Table 3, varies with the temperature up to 500 °C, whereas it remains almost unchanged between 600 and 700 °C. Differently from what observed for alkali lignin [15], LRR produces a huge amount of $CO_2$ that accounts for about 68.9 wt.% in the gas produced at 300 °C and decreases up to 59.2 wt.% in the gas produced at 700 °C. CO is produced in lower amount at its concentration remains almost constant in the range 29.2–31.7 wt.% at increasing temperature. The ratio between $CO_2$ and CO concentration in the gas produced at 700 °C is lower than the one observed for olive branches, kiwi branches and wheat straw processed in the same conditions as in this study characterized by a lower lignin and ash content. On the contrary, it is comparable to the ones obtained for pine bark and rice husks characterized by a higher content of lignin and ash, respectively.

**Table 3.** Composition and HHV value of the gaseous product from pyrolysis of lignin rich residue (LRR) at different final temperatures.

| T | °C | 300 | 400 | 500 | 600 | 700 |
|---|---|---|---|---|---|---|
| $CH_4$ | | 0.0 | 2.0 | 5.0 | 6.6 | 5.9 |
| CO | | 31.1 | 29.2 | 30.2 | 31.7 | 31.5 |
| $CO_2$ | wt.% | 68.9 | 68.8 | 63.2 | 59.2 | 58.9 |
| $C_2H_4$ | | 0.0 | 0.0 | 0.3 | 0.3 | 0.3 |
| $C_2H_6$ | | 0.0 | 0.0 | 1.1 | 1.1 | 0.7 |
| $H_2$ | | 0.0 | 0.0 | 0.2 | 1.1 | 2.7 |
| HHV | MJ/kg | 3.1 | 4.0 | 6.9 | 9.2 | 10.7 |

The first peak in the $CH_4$ release-rate curve (Figure 4) can be considered a marker of the presence of lignin [15,17], as it is higher with respect to the second one and correlates to the high lignin content of LRR.

$CH_4$ concentration reaches a maximum at 600 °C, whereas $H_2$ is produced at higher temperatures and its concentration increases with the temperature. C2 species are present as minor species since their concentrations attain a maximum of 1.1 wt.% at about 600 °C.

### 3.2.2. Char Characteristics

Table 4 reports relevant information on the bulk and surface chemistry of the chars produced at different temperatures as determined by elemental and proximate analyses, inorganic phase composition determination and pH measurement.

**Table 4.** Composition and properties of char of lignin-rich residue (LRR) obtained at different final temperatures. Percentage error is reported in brackets.

| Temperature | °C | 300 | 400 | 500 | 600 | 700 |
|---|---|---|---|---|---|---|
| H/C | | 0.940 (0.3) | 0.604 (0.3) | 0.415 (0.7) | 0.292 (0.5) | 0.151 (0.6) |
| O/C | | 0.300 (0.2) | 0.178 (0.4) | 0.070 (0.7) | 0.064 (0.3) | 0.047 (0.6) |
| Volatiles | | 48.6 (0.5) | 28.1 (0.4) | 18.0 (0.3) | 14.1 (0.2) | 11.0 (0.1) |
| Fixed Carbon | wt. % db | 38.7 (0.6) | 54.1 (0.4) | 60.7 (0.4) | 63.7 (0.2) | 66.5 (0.2) |
| Ash | | 12.8 (0.1) | 17.7 (0.1) | 21.3 (0.3) | 22.2 (0.3) | 22.6 (0.2) |
| Na | | 257 (1.2) | 350 (1.0) | 400 (0.9) | 460 (1.0) | 494 (0.8) |
| K | | 11710 (0.2) | 17000 (0.2) | 17500 (0.2) | 18980 (0.1) | 20390 (0.2) |
| Ca | mg/Kg | 2921 (0.4) | 4500 (0.3) | 4700 (0.2) | 4911 (0.2) | 5344 (0.2) |
| Mg | | 457 (0.8) | 650 (0.7) | 750 (0.6) | 760 (0.6) | 808 (0.6) |
| P | | 1349 (0.5) | 2000 (0.4) | 2200 (0.3) | 2441 (0.3) | 2673 (0.3) |
| pH | | 7.8 (0.3) | 9.9 (0.1) | 9.9 (0.1) | 10.0 (0.1) | 10.1 (0.1) |
| BET | $m^2$/gr | 2 (2.8) | 5.5 (2.5) | 126.7 (1.2) | 289.6 (0.3) | 124.3 (0.8) |

In the Figure 5 the effect of temperature on a specific characteristic is reported as the absolute value of the percentage variation of a specific characteristic for a temperature change of 100 °C, in order to highlight the range where pyrolysis temperature has a more

relevant effect. The percentage variation of the investigated characteristics, except for H/C and O/C ratio, has its maximum between 300 and 400 °C. As observed from TG analysis (c.f. Figure 2), most of the devolatilization occurs in this temperature range, as well as the highest release of CO and $CO_2$ (c.f. Figure 4), thus determining the significant reduction in H/C, O/C and volatile content.

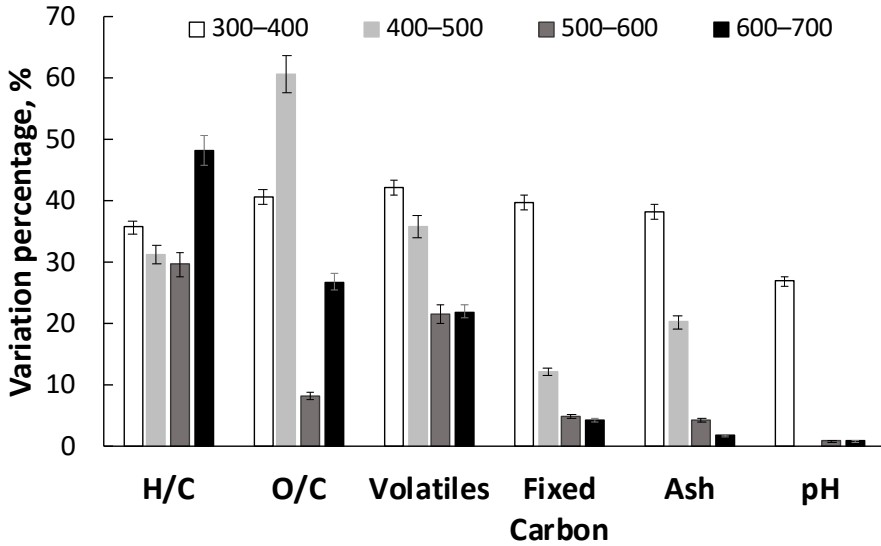

**Figure 5.** Absolute value of the percentage variation in LRR char characteristics for a temperature change of 100 °C.

Accordingly, the highest increase in fixed carbon was observed in the same temperature range. At these low temperatures, most of the inorganics are retained in the char and the variations in their concentration is very sensitive to the temperature change. It is likely that the high variation in char pH in this region is due to the cleavage of the carboxylic and carbonylic acidic groups forming, respectively, $CO_2$ and CO. Accordingly, above 400 °C pyrolysis temperature seems to have no effect on char pH.

Between 400 and 500 °C, the percentage variation of O/C attains its maximum, however the corresponding variations of H/C and volatiles content are lower than the ones observed between 300 and 400 °C. This result suggests that at higher temperatures, lighter volatiles with high molar O/C ratio are released. Additionally, in this temperature range, the variation in ash content with the temperature still high indicates a retention of the main inorganics in the chars.

The percentage variation of the volatiles content decreased at temperature higher than 500 °C, but is still high up to 700 °C. At the same time O/C ratio varies only slightly between 500 and 600 °C, whereas a significant variation of H/C ratio is observed attaining its maximum between 600 and 700 °C. This result is consistent with the significant release of $CH_4$ and $H_2$ observed above 500 °C (c.f. Figure 2). The change in volatile content above 500 °C affects, at the same time, fixed carbon and ash content, thus producing only slight variation of these two characteristics (c.f. Table 4).

In Table 4, the temperature effect on the char porosity is reported. Char produced at 300 °C has still a compact structure, whereas, as temperature increases, the porosity develops to BET surface values similar to the ones typical chars obtained from woody and herbaceous biomass [21]. It is worth noting that chars obtained from steam-assisted pyrolysis of commercial alkali lignin conducted in the same reactor under similar thermal conditions, showed a compact structure even at high pyrolysis temperature [18]. The difference could be ascribed to the higher content of volatiles in the LRR (c.f. Table 1) with respect to the alkali lignin [22], but also to the presence of a rigid structure not significantly altered by the enzymatic hydrolysis pretreatment of biomass undergoing fermentation process. The images of LRR raw material and char produced at 600°C at different magnification are presented

in Figure 6. As a comparison, also in Figure 6, the SEM analysis of alkali lignin and the corresponding char produced at 600 °C [17] are reported. SEM micrographs confirmed that the high temperature char obtained from steam-assisted pyrolysis of alkali lignin carried out in the same reactor under similar thermal conditions has a compact structure. LRR structure still has evident cellulose fibers which are preserved after the thermal treatment. The difference between the char from LRR and alkali lignin char could be ascribed to the presence of a rigid structure in alkali lignin not significantly altered by the enzymatic hydrolysis pre-treatment of biomass undergoing fermentation process.

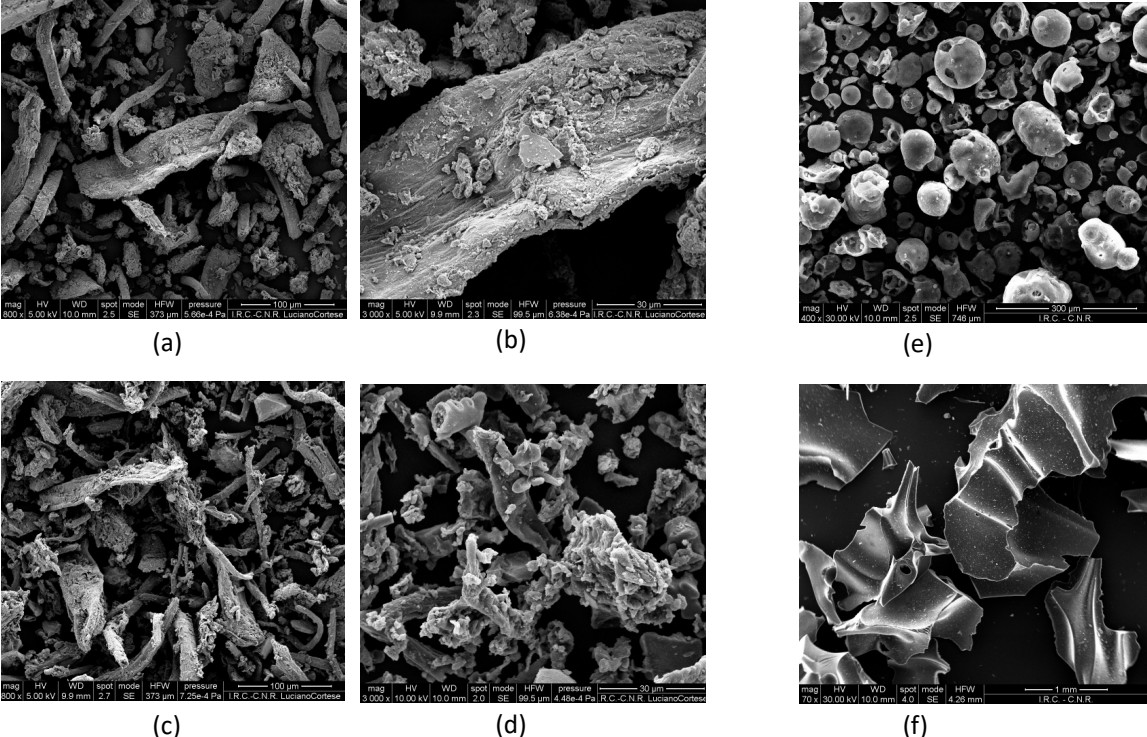

**Figure 6.** SEM analysis LRR (**a**,**b**), LRR char at 600°C (**c**,**d**); alkali lignin (**e**), alkali lignin char, (**f**) at 600 °C adapted from [19].

## 4. Conclusions

In this study, lignin-rich residue was explored at different temperatures in slow pyrolysis conditions and the results were compared with those reported in a previous work on the corresponding raw biomass (Around donax), and the alkali lignin. The main results can be summarized as follows:

- LRR shows a different thermal evolution compared to alkali lignin. Contrary to alkali lignin, LRR is subject to a prompt weight loss in the range of 280–340 °C due to the presence of holecellulose residue.
- Regarding the products' distribution and composition, differently from alkali lignin, the effect of temperature was mostly evident between 300–400 °C. A great reduction in the char yields is observed up to 400 °C, where significant reduction in H/C and O/C ratios as well as of volatile content in the chars were observed. At temperatures higher than 500 °C a significant reduction in H/C ratio and volatile compounds was observed consistently to the $CH_4$ and $H_2$ release observed above 500 °C. The presence of holocellulose in LRR was also responsible for the increased production of liquid-products along with a reduction in char yield above 400°C. On the contrary, alkali lignin produced comparable amounts of liquid and char even at the highest temperature. Gas production increased up to 500 °C and does not vary significantly at higher temperatures, in terms of both yield and composition.

- The temperature significantly impacted the porosity of the resulting char. The char structure developed with the temperature, starting from a compact structure at low process temperature and reaching the maximum BET value at about 600 °C. On the contrary, the char produced from alkali lignin features a compact structure even at high temperature. The differences could be ascribed to the high content of volatiles in the LRR. The SEM micrographs confirmed this differences showing a compact structure of LRR char with evident cellulose fibers, preserved after the thermal treatment, with respect to the alkali lignin.

**Author Contributions:** Conceptualization, C.M.G. and P.G.; methodology, C.M.G. and P.G.; validation, C.M.G. and P.G.; formal analysis, C.M.G. and F.S.; investigation, C.M.G.; writing—original draft preparation, C.M.G. and P.G.; writing—review and editing, C.M.G., P.G. and R.R.; supervision, C.M.G., P.G. and R.R. All authors have read and agreed to the published version of the manuscript.

**Funding:** This research received no external funding.

**Institutional Review Board Statement:** Not applicable.

**Informed Consent Statement:** Informed consent was obtained from all subjects involved in the study.

**Data Availability Statement:** Not applicable.

**Conflicts of Interest:** The authors declare no conflict of interest.

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
