# Peer review of "Influence of Pyrolysis Temperature on Biochar Produced from Lignin–Rich Biorefinery Residue"

_2305-7084, doi:10.3390/chemengineering6050076_

Round 1

Reviewer 1 Report

The study has been thoroughly conducted, with results that are sound and well-detailed. Somme comments and corrections can be found in the annexed PDF. Also:

1. Lines 56-66: The authors give some references concerning the pyrolysis studies conducted on LRR. However, details are missing. Can the authors further elaborate the results and consequent conclusions found in literature concerning this process?

2: The conclusion needs to be fleshed out more by adding more details from the results obtained.

Reviewer 2 Report

This work os about the effect of the temperatura on the production of deverão produts in a pyrolysis process.

The article is interesting and present some innovative results. 

Nevertheless, I proposed to study the effect of the pressure in this system, using the best operating conditions achieved on this work, to improve the paper.

Reviewer 3 Report

This paper, entitled Influence of Pyrolysis Temperature on Biochar Produced from Lignin–rich Biorefinery Residue, is a scholarly work and can increase knowledge on this domain. The content is relevant to ChemEngineering and the study is original. The manuscript is quite well written and well related to existing literature.

I have some general and specific comments: - Why working with slow pyrolysis conditions (5°C per min)? In this condition, volatilization of cellulose could be obtained before pyrolysis (volatilization between 180 to 225°C). With such ratio, the sample stays some time at the temperature of volatilization and some compounds could be lost. - How many samples were treated and characterized? How many replication? - In Figure 2, is it average for each curve or one sample? Please give accuracy or standard deviation. - Please provide accuracy of data in Table 1 and Table 2. Same comments for Table 3 and Table 4. - Please provide error bars in Figure 5. - There's no Figure 6 and SEM are shown in Figure 7. Maybe there's one figure missing or this is a mistake. - How many samples were observed for SEM analysis? How many replication? - Is there any measurement of cationic exchange capacity or water retention of such biochar? - What could be the application of such biochar obtained by this way? Plese discuss about potentialities of such product. - These experiments were carried out at labscale, is there any experiment scheduled at highest scale? What about transfer or applicability at pilot scale or real scale? - Please discuss about mass balance and energy balance.   As it, the manuscript is not fully acceptable for publication and requires some amendments and modification, or additional information.

Round 2

Reviewer 3 Report

The authors provide a revised version of their manuscript taking into account all the comments and requests of amendments. The authors provide detailed answers to all the comments. I agree with all the answers and the revised version. As it, the manuscript is now fully acceptable for publication.